# Will Policy Uncertainty Deteriorate Haze Pollution? A Spatial Spillover Perspective

**DOI:** 10.3390/ijerph181910229

**Published:** 2021-09-28

**Authors:** Xiulin Qi, Xin Wang, Xiao Jin, Zhenyu M. Wang, Beibei Zhang, Chuanhao Wen

**Affiliations:** 1Business School, Zhengzhou University, Zhengzhou 450001, China; qixiulin@zzu.edu.cn; 2Department of Economics, Southwestern University of Finance and Economics, Chengdu 610074, China; bashuwang@163.com; 3School of Public Policy & Management, Tsinghua University, Beijing 100084, China; jin-x18@mails.tsinghua.edu.cn; 4School of Social Sciences, University of Macau, Macau 999078, China; wzhenyu@pku.edu.cn; 5School of Management and Economics, Beijing Institute of Technology, Beijing 100081, China; beibeizhang9533@163.com; 6School of Economics, Yunnan University, Kunming 650091, China

**Keywords:** policy uncertainty, haze pollution, spatial spillover effect

## Abstract

Haze has been a severe problem in China for some time, jeopardizing air quality, public health and sustainable growth. This paper examines the direct effect and spatial spillover effect of policy uncertainty on haze pollution with a spatial panel model, using prefecture-level data from 2004 to 2016. This study shows that: (1) policy uncertainty has increased the level of local haze pollution and has a significant spatial spillover effect on surrounding areas; (2) although local policy uncertainty has increased the haze pollution in geographically adjacent cities, it only affects the cities within the province with similar economic distances; and (3) the policy at the central level can effectively alleviate the impact of policy uncertainty at the local level on haze pollution, especially in relation to the spatial spillover effect, but still has limitations in eliminating the direct effect, which is due to the ineradicable nature of policy uncertainty.

## 1. Introduction

China has experienced rapid economic growth in recent decades, but this growth has also damaged the environment. In recent years, haze pollution has attracted great public attention because it has a large impact on the public’s daily lives. The central government has focused strongly on haze pollution control and introduced a great number of environmental protection policies, including introducing haze pollution control performance into the performance evaluation system of local officials [1]. Haze pollution is still a widespread and major issue in China, even since the promulgation of numerous environmental policies. Existing studies find that some local governments mainly take measures to control air pollution and create “political blue skies” temporarily [2]. The Ministry of Ecology and Environment of China reported that more than half of prefecture-level cities (180 cities) were labeled as having poor air quality in 2019. More specifically, it was recorded that 1666 days of heavy haze pollution and 452 days of extremely severe pollution occurred across 337 cities. Fine particulate matter (PM_2.5_), as the primary pollutant, accounted for 78.8% of the days with heavy or even more significant levels of pollution [3].

Although haze is related to meteorological conditions, it is essentially due to the unsustainable mode of development in China. Economic activities in the early stages of development will generate more pollution, but when the economy develops to a certain level, the level of local pollution will decline with the continuously increasing investment in environmental protection and the gradual conversion of low-end industries. The environmental Kuznets curve indicates an “inverted U-shaped” relationship between environmental pollution and economic development [4]. Chen et al. [5] find that an approximately inverse-U-shaped relationship between haze pollution and economic growth exists in most provinces in China and that haze pollution is expected to intensify with economic growth in most provinces. Considering the fierce competition among local officials of the Chinese government in relation to the Gross Domestic Product (GDP) and the dominant position of local governments in the transformation of industrial structure and pollution control, it is worth exploring the political-economic reasons behind the regional differences in China’s haze pollution [6,7,8,9]. Studies suggest that local governments can play an active role in haze governance, but that the environmental outcomes are not promising. One possible reason for this is that the targets of local governments and the central government are not exactly consistent. Local government officials in promotion tournaments have strong incentives to develop the economy, despite the cost for the environment. This is supported by empirical evidence which indicates that regions’ pollution levels are positively correlated with the number of officials promoted [10]. In addition, the government and enterprises may engage in collusion so as to increase both economic growth rates and tax revenue at the cost of higher pollution [11,12].

However, the extant literature mainly focuses on the static relationship between local governments and haze pollution, ignoring that government behavior can be dynamic—especially in terms of the effect of policy uncertainty on haze pollution. Effective control of haze requires a sustained and stable policy environment. However, local governments often fail to maintain a stable policy for a long period, which introduces some degree of policy uncertainty. Theoretically, the uncertainty will affect decision-making and the implementation of haze control governance, which will ultimately be reflected in the differentiated outcomes of haze governance in local governments. Zhang and Tang [13] indicate that the policy uncertainty caused by official turnover aggravates local haze pollution, but their work does not take the spatial spillover characteristics of haze pollution into consideration, which may lead to biased estimations. Guo and Shi [14] find that official turnover, a politically sensitive period, creates a deterrent effect on collusion. Furthermore, they find only a short-term effect on the improvement of air quality caused by official turnover, with haze pollution levels returning to normal conditions when the politically sensitive period passes. Hence, the effect of policy uncertainty on haze pollution is still unclear. In addition, although many studies have applied the spatial Durbin model in investigating the effect of spatial spillover on haze pollution, more thorough analysis and explanation are needed. First, the existing research concerned only the spread of haze pollution in geographical space but ignored other possible paths of diffusion. Second, existing studies have failed to effectively identify the mechanisms of the spatial spillover effect of haze pollution.

Based on prefecture-level data from 2004 to 2016, this paper examines the direct effect and spatial spillover effect of policy uncertainty on haze pollution with a spatial panel model. This study finds that policy uncertainty will significantly deteriorate haze pollution levels locally as well as in nearby cities. The mechanisms include geographical diffusion and local government environmental competition. After enactment of the strictest policy, the Assessment Method for the Implementation of the “Measures for the Assessment of the Implementation of the Air Pollution Prevention and Control Action Plan (Trial)”, was promulgated, the effect of policy uncertainty on local haze pollution decreased and there was no spillover effect. This shows that the central policy can effectively curb the impact of policy uncertainty on haze pollution at the local level.

The main contributions of this article are as follows. First, this research complements the limited existing studies that focus on how policy uncertainty affects haze pollution and its spatial spillover effect. Considering the transferability of haze pollution and the competition among local officials over GDP growth, it is crucial to explore the spatial spillover effect of policy uncertainty. This work may help to improve the accuracy of estimates and provide a comprehensive understanding of the effects of policy uncertainty. Second, this study investigates the mechanisms of the spatial spillover effect. This paper divides haze pollution spatial spillover mechanisms into two categories: geographic distance and economic distance. Moreover, they are identified by constructing the “intra-province space matrix” and “inter-province space matrix”. In particular, this paper is the first empirical study to explore the logic chain of “local policy uncertainty—local environmental pollution—local environment “race to the bottom”—environmental pollution in competitive areas”. Last, this paper finds that the effects of policy uncertainty on haze pollution remain after the introduction of strict and unified environmental regulation by the central government.

The remainder of this paper is structured as follows: in the second section, this paper reviews the literature regarding policy uncertainty and provide key hypotheses. The third section introduces our data and empirical methodology, followed by the demonstration of the empirical results in the fourth section. The paper concludes with the research findings in the fifth section.

## 2. Literature and Hypotheses

### 2.1. Official Turnover, Policy Uncertainty and Local Haze Pollution

Various studies have shown that the turnover of government officials, which will lead to policy instability, is an appropriate proxy variable for policy uncertainty [15]. It is more appropriate to take the turnover of local officials as the proxy variable of policy uncertainty in China due to the unique nature of the country’s economic decentralization and political centralization. The reason is that China’s unique decentralization system imparts authority to local officials over land, policies and credit. Chinese local officials exert heavier political influence over city governance than their counterparts in Western countries and, therefore, turnover also has a greater impact on policy stability. Empirical studies show that policy uncertainty caused by official turnover will not only affect macroeconomic growth [16], but also affect the decision-making and behavior of enterprises at the micro level [17].

The impact of policy uncertainty on haze pollution is theoretically ambiguous and empirically controversial.

On the one hand, policy uncertainty will exacerbate haze pollution. Although the frequent occurrence of haze weather is related to meteorological conditions, it essentially stems from the unreasonable development mode. Under China’s unique decentralization system, local governments are mainly responsible for pollution control governance. Although there is some controversy, it is generally believed that fiscal decentralization has aggravated China’s environmental pollution [18]. As the “agents” of the central government, local governments often conduct strategic actions in response to external constraints, including, in particular, command by higher authorities. For example, Shi et al. [19] find that the air quality improves significantly during the “Two Sessions” period, which is mainly reflected in the perceivable indices such asPM_2.5_, PM_10_ (particulate matter with particle size below 10 microns) and SO_2_. However, the air quality deteriorates dramatically after the “Two Sessions”. The degree of deterioration is more severe than the degree of improvement during the “Two Sessions”. Chen et al. [20] indicate that within one month of the end of the Olympic Games, the pollution index of Beijing began to rise rapidly. Liang and Laura [21] point out that, among many pollutants, if their visibility to the public is high and they are in the assessment target system, the governance effect will be significant, otherwise the effect will not be significant. This strategic behavior reflects the “failure” of local governments in the treatment of air pollution, and this can be attributed to two reasons. First, because China’s local officials are facing tremendous pressure for promotion, there is an incentive to sacrifice the environment in exchange for economic growth under the assessment system that pays more attention to economic growth. At the macro-level, there are more promoted officials in polluted areas [22]. At the micro-level, with collusion between the government and enterprises, officials can achieve higher economic growth and more taxes at the cost of deregulating pollution [23]. Second, because of the frequent spatial spillover of pollution, there is “free-riding behavior” in governance [8].

Under China’s economic decentralization and political centralization system, policy uncertainty may exacerbate haze pollution. First, policy uncertainty means more short-sightedness as it results in trading the environmental impact for more growth. Expanding local investment is an important way for officials to accumulate promotion capital in China, where local officials compete fiercely over GDP growth and fiscal revenue. Since it takes some time for investments to show effects, officials will adjust relevant policies to pursue larger investments from the very beginning of their tenure under limited-term constraints. Studies have shown that the turnover of local officials will increase the overall investment growth rate in China [24]. In the long run, environmental pollution, such as haze, will restrict economic growth through mechanisms, such as reducing the attractiveness of cities and slowing the accumulation of human capital. However, in the short run, economic development can be achieved at the cost of the environment. Therefore, since economic growth plays a more important role in officials’ performance evaluation than environmental protection, a higher level of policy uncertainty will expand investment in the short term, thereby exacerbating haze pollution. Second, policy uncertainty creates a period of regulatory and accountability gaps. At present, haze control has been included in the performance evaluation system for local officials, and emission behaviors are strictly regulated by the government. However, during the regulatory gap period and the responsibility gap period accompanied by policy adjustment, polluters who emit haze-causing pollutants are not held accountable without regulation, which leads to increased haze pollution. In addition, if the new officials’ working style is different from that of his/her predecessors, the duration and the impact intensity of supervision and the responsibility gap periods will be further amplified.

On the other hand, some studies suggest that policy uncertainty may also curb haze pollution. First, policy uncertainty slows down business investment. According to the view of investment information, uncertainty increases the benefit of waiting for new information, and thus enterprises will stop investing in the current period until the cost of postponing investment exceeds the benefit of waiting. Pastor and Veronesi [25] construct a theoretical model to depict enterprises’ investment reduction in the face of uncertainty. When companies slow investment, they reduce their emissions, and this contributes to the reduction in local levels of haze. Second, policy uncertainty will reduce the collusion between the government and enterprises at the local level, thereby increasing the deterrent effect on polluters. The excessive emission of haze pollutants is often the result of collusion between the government and enterprises. In a collusive relationship; companies cut production costs through heavy emissions with the tacit approval of local officials who use this to accumulate political assets for promotion and gain additional power. However, the political connections of local companies are usually limited to local officials. Once local officials have left office, the old collusion relationship breaks down, meaning that policy uncertainty will result in companies lowering their emission levels. Fredrikssona and Svensson [26] find that, where corruption is a very serious problem, the turnover of officials leads to the strengthening of environmental regulation policies and the reduction of local pollution levels. Gao and Liang [27] also argue that official turnover can suppress environmental pollution by breaking the existing network of collusion between the government and enterprises, but has only limited and temporary effects in mitigating pollution. Based on the above analysis, this paper proposes the following hypotheses:

**Hypothesis** **1** **(H1a).**
*Policy uncertainty will deteriorate local haze pollution.*


**Hypothesis** **1** **(H1b).**
*Policy uncertainty will curb local haze pollution.*


### 2.2. Policy Uncertainty Affects the Spatial Spillover Effect of Haze Pollution

According to Waldo Tobler’s first law of geography, “everything is related to everything else, but near things are more related than distant things.” Haze pollution has an obvious spatial diffusion and transferring effect, which has been fully verified in previous studies [28,29]. The spatial correlation of haze pollution first comes from its natural attributes. It will be transmitted to the surrounding areas geographically, with changes in the wind direction and humidity between regions. However, the spatial correlation may come from the behaviors of local governments under the constraints of competition. To attract new enterprises and avoid capital moving to areas with less stringent environmental controls, local governments may relax environmental controls and engage in an environmental “race to the bottom” [30].

Combined with China’s unique decentralized system and the intense promotion tournaments among local officials, the environmental “race-to the bottom” is even more acute. The logic is that once other competitors have achieved short-term economic growth at the expense of polluting the environment, local officials have a strong incentive to follow suit to avoid being outcompeted. The existence of an environmental “race to the top” means that smog disperses spatially in a different, non-geographical way; local smog pollution will be “transported” to the areas where officials are competing with local officials, thereby aggravating their smog pollution. In other words, if policy uncertainty in one place affects local levels of haze pollution, then the spatial spillover effect of such an impact can occur, not only in geographically adjacent areas, but also among areas where local officials are competing with. Based on the above analysis, this paper proposes two hypotheses on the mechanisms of the spatial spillover effect from haze pollution:

**Hypothesis** **2** **(H2).**
*The impact of policy uncertainty on haze pollution has a spatial spillover effect in adjacent geographical areas.*


**Hypothesis** **3** **(H3).**
*The impact of policy uncertainty on haze pollution has a spatial spillover effect among the regions that officials are competing with.*


## 3. Research Design

### 3.1. Econometric Model

According to the above analysis, policy uncertainty may affect local haze pollution and have a spatial spillover effect. Therefore, this paper applies the spatial Durbin model. This model has the following advantages. First, the spatial Durbin model is a standard framework for capturing various spatial effects and can be transformed into a space hysteresis model and a space error model under different coefficient settings [31]. Second, regardless of whether the real data generation process is spatial lag mode or spatial error mode, the spatial Durbin model can guarantee an unbiased estimation coefficient. Third, the spatial Durbin model has no limitation on the scale of spatial spillover effect. The estimation equation takes the following form:(1)$$\ln PM_{2.5it}=\theta +\rho W\ln PM_{2.5it}+\alpha uncertainty_{it}+\beta W*uncertainty_{it}+\gamma X_{it}+\eta W*X_{it}+\mu_{i}+\lambda_{t}+\varepsilon_{it}$$

The dependent variable $\ln PM_{2.5it}$ is the log transformation of the haze pollution level. The key independent variable $uncertainty_{it}$ measures the policy uncertainty level; $X_{it}$ consists of a set of control variables capturing regional economic development and environmental regulation which are discussed in detail in Section 3.3.3; $u_{i}$ and $\lambda_{i}\;$ represent city fixed effect and time effect respectively; $\varepsilon_{it}$ represents the error term; W means the spatial weight matrix; and α and β measure the direct effect and spatial spillover effect of policy uncertainty on haze pollution.

### 3.2. Data

The PM_2.5_ concentration data used in this paper comes from the Socioeconomic Data and Applications Center (SEDAC) which is affiliated with the Center for International Earth Science Information Network (CIESIN) at Columbia University. The PM_2.5_ concentration data from SEDAC is consistent with the data released by the Ministry of Ecology and Environment of China. This data is highly reliable and is widely used in haze pollution-related research. CIESIN transforms the global PM_2.5_ data in the form of grid data based on the aerosol optical thickness (AOD), measured by moderate-resolution imaging spectroradiometer (MODIS) and multi-angle imaging spectroradiometer (MISR). This article then uses ArcGIS to analyze the data and obtain the annual PM_2.5_ concentration data for China’s prefecture-level cities.

There are two other measurements of PM_2.5_ that were used in previous studies. One method is to incorporate satellite data and ground-based monitoring data into the two-stage spatial statistical model, and the other is to calculate the PM_2.5_ concentration of previous years based on the ratio of SO_2_ emissions to PM_2.5_ concentration in China’s Environmental Annual Report. Compared with these two measurements, the PM_2.5_ concentration data this paper uses is less influenced by different algorithms but is not as precise as ground-based monitoring data. However, considering that this paper examines the regional differences in haze pollution, ground-based monitoring data obtained from specific monitoring stations cannot accurately represent the pollution level of the entire region.

To build the indicator of policy uncertainty, there are three steps. First, this paper obtained information on the current mayor and secretary of the Municipal Party Committee of the prefecture-level city from websites such as People.com.cn (accessed on 10 September 2020). The data on the mayor and secretary’s predecessor on the Baidu website were then downloaded. Finally, it was necessary to retrieve historical news to improve the validity of the data. For missing information, this paper referred to open databases, such as the China Stock Market & Accounting Research Database and the prefecture-level city party secretary database.

Control variables were collected from the Statistical Yearbook of Chinese Cities and the Statistical Yearbook of Chinese Urban Construction. After combining the data from different sources and excluding missing values and outliers, the unbalanced panel data, spanning from 2004 to 2016 and with a sample size of 3731, was finally obtained.

### 3.3. Variables

#### 3.3.1. PM_2.5_

The main pollutant that forms haze is PM_2.5_, which refers to a particle with a diameter of less than or equal to 2.5 microns. PM_2.5_ contributes to the formation of haze pollution, and haze pollution can further aggravate the accumulation of PM_2.5_. With high humidity in the air under hazy weather, some polluting gases generate PM_2.5_ through atmospheric chemical reactions. The accumulation of PM_2.5_ also accelerates the generation of haze. PM_2.5_ can be absorbed by lungs and enter the blood, and thus can be very harmful to the human body. Long-term inhalation of PM_2.5_ may lead to cardiovascular and respiratory diseases and lung cancer. This paper follows the common practice used in existing studies and takes PM_2.5_ annual concentration data as the proxy variable for the haze pollution level.

#### 3.3.2. Uncertainty

Three main indicators are used to measure policy uncertainty in the existing literature. Baker and Nicholas [32] suggest constructing an economic uncertainty index according to the number of relevant news reports. The second indicator comes from the World Uncertainty Index (WUI) [33]. The third indicator uses the turnover of officials as a proxy variable for policy uncertainty. This paper uses the third measurement because local governments play important roles in economic development in China. In addition, economic uncertainty alone is not enough to cover this environmental problem. A more comprehensive index to measure policy uncertainty is needed. In addition, differing from the previous measurements of policy uncertainty, this paper constructs the level of policy uncertainty by the sum of the turnover number of Communist Party of China secretaries (party secretaries) and mayors. The reasons are as follows. First, although party secretaries have more political power than mayors, mayors are often in charge of environmental governance. Second, different local party secretaries often have different ways of distributing the actual work. Thus, only taking the turnover of party secretaries as a measurement may underestimate the impact of policy uncertainty on haze pollution.

#### 3.3.3. Control Variables

The set of control variables includes regional economic development, such as GDP level, industrial structure, degree of openness, resource dependence, capital investment, urbanization, population density and environmental regulation.

GDP: The level of economic development is closely related to haze pollution. Economic development often means more resources, investments and pollution emissions; meanwhile, a higher level of economic development will promote local investment in environmental protection or the shutting down of polluting factories to curb haze pollution. This study applies the measurement of the regional development level with the regional GDP, and uses the GDP deflator to eliminate the impact of price fluctuation.

env: Effective environmental regulation can force enterprises to develop green technology innovation by changing their production mode to curb haze pollution. This paper measures the environmental regulatory intensity in various regions by the share of investment in environmentally friendly facilities in the GDP.

ind: Different industries emit different amount of haze pollutants and have different impacts on the environment. This paper measures the industrial structure by the share of the secondary industry in the GDP.

urban: Urbanization leads to pollution because high levels of motor vehicles and factory emissions make haze pollution in cities more serious than in rural areas. This paper measured the urbanization level by the proportion of the non-agricultural population in the total population.

pden: Higher population density leads to pollution too. This paper measures the population density by the population per square kilometer. High population density implies more human activities and leads to higher pressure on the environment.

open: The degree of openness can also affect the haze pollution level. This paper measures the degree of openness by calculating the proportion of foreign direct investment (FDI) in GDP. There are two arguments on how foreign investment affects environmental pollution. One is the “pollution paradise hypothesis”, which holds that developing countries will lower the threshold of environmental access in order to attract foreign investment, and thus causing environmental pollution. The other is the “pollution halo hypothesis”, which holds that foreign investment will lead to more environmentally friendly technologies, thereby improving the environmental quality in host countries.

res: The exploitation of natural resources can also be related to pollution. If local development is dependent on local mineral resources, then the development is more likely to be positively correlated with pollution emissions. In this paper, the proportion of extractive industries in total employment is used to measure the degree of resource dependence.

mater: This paper measures the physical capital per labor by the ratio of the physical capital stock over the total number of employees.

## 4. Results

### 4.1. Spatial Correlation Test

Before construction of the spatial econometric model, a test was performed to verify the spatial autocorrelation of the data. By constructing the reciprocal matrix of distance, this paper utilizes Moran’s I index to measure the spatial autocorrelation characteristics of PM_2.5_ from 2004 to 2016 in China. The range of this index is [−1, 1]. If Moran’s I index is close to 1, it represents the positive spatial autocorrelation of cross-regional haze pollution. If it is close to -1, it represents the negative spatial autocorrelation. If it is close to 0 or not statistically significant, there is no spatial correlation.

Table 1 shows that haze pollution has a positive and statistically significant spatial correlation, and Moran’s I index is concentrated in the range of 0.1–0.2 over the 13 years. Therefore, it is necessary to use the spatial econometric model to analyze the impact of policy uncertainty on haze pollution.

### 4.2. Spatial Panel Estimation Results

To introduce spatial variables into our econometric framework, this paper applies the spatial Durbin model, which is a generalized spatial model compared with the spatial lag model and the spatial error model [31,34]. The Durbin model can estimate the direct effect, the spatial spillover effect and the total effect.

The first three columns in Table 2 report the estimation results using the reciprocal matrix of distance as the spatial weight matrix. Column 1 shows that policy uncertainty has a positive direct effect on local haze pollution, and the result is statistically significant at the 1% level. This verifies hypothesis H1a, which indicates that policy uncertainty will promote local haze pollution through mechanisms of regulatory gaps and responsibility gaps. This promotion effect is greater than the restraining effect of policy uncertainty on haze pollution, and thus the net effect is positive.

Column 2 reports the spatial spillover effect. Local policy uncertainty has a positive impact on haze pollution in neighboring cities, and this impact is statistically significant at the 5% level. The spatial spillover effect is very large. To ensure the robustness of the conclusion, this paper further uses the adjacency matrix instead of the reciprocal distance matrix to re-estimate and finds that the magnitude and significance of the coefficients remain unchanged. Overall, the spatial spillover effect accounts for most of the total effects in the model.

Table 2 also reports the effects of control variables on haze pollution. Economic development has reduced local haze pollution. The increase in environmental protection investment and the transfer of low-end industries accompanying economic growth have curbed haze pollution, which is consistent with existing findings [5]. The higher the level of urbanization is, the higher the level of smog pollution is, as a large number of factories and motor vehicles are concentrated in cities, thereby generating more air pollutant emissions. The greater the population density is, the higher the level of haze pollution is, which means that haze pollution is closely related to human pollution. The increase in resource dependence and physical capital per labor has significantly increased the level of haze pollution, which shows that the development of mineral resources and large-scale capital investment will increase haze pollution. The degree of openness will not affect the level of smog pollution, possibly because the mechanisms based on the pollution paradise hypothesis and the pollution halo hypothesis coexist and their effects cancel each other out in China.

### 4.3. Space Overflow Mechanisms

After the spatial spillover effect of haze pollution due to policy uncertainty is uncovered, the next question lies in identifying the mechanisms through which spatial spillover occurs. According to the existing literature, haze pollution can be diffused through geographical mechanisms, such as atmospheric circulation, or non-geographical mechanisms, such as a regional race to the bottom in emissions to raise GDP output. However, it is a major challenge to separate the above two mechanisms empirically. Fortunately, the unique performance review system for Chinese officials, “one-level-down management”, makes it possible to differentiate the two mechanisms. Under this arrangement, the competitors of prefecture-level city officials are confined to those both within the same province and with a similar economic development level [35]. Therefore, this paper can identify the above two mechanisms by constructing the “inter-provincial spatial matrix” and the “intra-provincial spatial matrix” using geographical and economic distances. Two main steps are involved in building the matrix. First, this paper computes the difference in the mean GDP output among the cities to establish an economic distance. The closer the mean GDP output is, the shorter the economic distance will be. Second, this paper follows Yu and Zhou [36] to build the “intra-provincial spatial matrix” and the “inter-provincial spatial matrix”. The former matrix gives weight to cities within the province according to geographical distance and economic distance, while the latter matrix gives weight only to cities across provinces.

After construction of the matrices, this paper proceeds to segregate geographical and non-geographical mechanisms. The identification logic is as follows. If the geographical mechanism dominates, air pollution diffusion would be observed in geologically adjacent cities regardless of whether they are within or outside of a province. In other words, the spillover effect will manifest both with an intra-provincial spatial matrix and inter-provincial spatial matrix based on geographical distance. If the non-geographical mechanism dominates, such as the race-to-the-bottom emission, the existence of the spatial spillover effect would concentrate only in cities with similar economic levels within the province. This is because intra-provincial officials are motivated to connive at emissions in the competition of GDP output under the one-level-down management system. At the same time, the spillover effect would be statistically insignificant in the inter-provincial matrix with economic distance because officials are not in the same ranking pool.

Table 3 and Table 4 summarize the findings based on the methodology above. The results in Table 3 indicate that when geographical distance is used for the spatial matrix, there is a significant spatial spillover effect in intra-provincial and inter-provincial cities. In other words, local policy uncertainty not only exacerbates the local haze pollution level, but also spills over the adjacent areas, which verifies Hypothesis H2. One noticeable observation from the coefficient is that the spatial spillover effect of haze pollution is more significant in inter-provincial cities than intra-provincial cities. One possible explanation is that province-level officials are accountable for air pollution control within their own province. Hence, they focus on pollution regulation in cities within their own province, which also includes minimizing spillover control of pollution across cities within the same province. In contrast, the negative externalities arising from pollution spillover to neighboring provinces are of secondary concern to province-level officials. Under such a system, it is rational to deduce that officials prefer to place high-emission industries at province borders and to carry more lenient regulations. As a result, the pollution spillover effect is magnified at the inter-provincial level. In summary, Table 3 provides empirical evidence for the “beggar-thy-neighbor” theory in research on air pollution control policy.

Table 4 shows that when the economic distance is used for the spatial matrix, policy uncertainty still deteriorates local haze pollution but the spillover effect is significant only in intra-provincial cities. This finding is consistent with our expectations which indicates that the regional race to the bottom in emissions would jeopardize air quality both locally and in cities with direct competition. Under the “one-level-down management” system, prefecture-level officials directly compete with cities of similar economic levels and within the same province. The industrial emission rise will trigger direct competitors to follow suit in fear of lagging behind in GDP output, leading to race-to-the-bottom emissions and heightening pollution spillover for intra-provincial cities. Performance competition is absent in officials in inter-provincial cities, hence there is little pollution spillover for inter-provincial cities with close economic distances. Table 4 serves as an indirect validation of Hypothesis H3.

### 4.4. Restraining Effect of Central Policy

The analysis in Section 4.3 shows that policy uncertainty will increase local haze pollution and lead to a spatial spillover effect because of the strategic behavior of local governments. In the current practices of haze control in China, the promulgation and implementation of relevant policies come from local and central governments. The central government has a higher level of authority. Therefore, a significant question about haze governance is whether policies from the central government can alleviate the impact of policy uncertainty at the local government level on haze pollution.

To understand the effectiveness of policies from the central government, this paper examines the effect of policy uncertainty before and after the release and implementation of the unified and strict measurement “Measures for the Assessment of the Implementation of the Air Pollution Prevention and Control Action Plan (Trial)”, which is the strictest system for atmospheric environment management responsibility and assessment.

In September 2013, the State Council issued the Notice on the Issuance of the Action Plan for The Prevention and Control of Haze Pollution, proposing that PM_2.5_ and PM_10_ indicators should be taken as binding indicators in order to establish a target-responsibility assessment system, which takes environmental quality improvement as the core. Six departments including the Ministry of Environmental Protection issued the “Measures for the Assessment of the Implementation of the Air Pollution Prevention and Control Action Plan (Trial)”, which clarified the evaluation content, evaluation method, evaluation procedure, and disciplinary measures of atmospheric pollution prevention and control.

The publication of these policies marks the establishment of the Chinese atmospheric environmental management system. According to the measures, the central government will increase support for regions with excellent assessment results. In contrast, local governments that fail to pass the assessment will obtain fewer funds from the central government. Moreover, they will be questioned by the Ministry of Environmental Protection together with organizations, such as supervisory departments.

Because the assessment method clearly states that performance will not be assessed in 2013, this paper divides the sample into two sub-samples with two time periods: 2004~2013 and 2014~2016. This paper separately investigates the effects of policy uncertainty on haze pollution before and after the implementation of the central level policy that includes “haze reduction”.

Panel A in Table 5 shows that the direct effect of policy uncertainty on haze pollution and the spatial spillover effect was positive and statistically significant before 2014. Although policy uncertainty still had a significant positive effect on local haze pollution after 2014, the spatial spillover effect and total effect are insignificant.

These results indicate that the assessment policy from the central government regulates the behavior of local officials with clearer assessment indicators and severe punishment measures. This regulation only marginally alleviates the impact of policy uncertainty on haze pollution. This result suggests that unifying the assessment alone may generate a very limited policy effect.

### 4.5. Robustness Tests

There may be endogeneity problems caused by missing variables and reverse causality in this paper. Although studies have shown that the spatial Durbin model has advantages in dealing with endogeneity problems caused by missing variables [31], a better way to address endogeneity problems is by utilizing appropriate instrumental variables. To tackle the endogeneity problem, this paper follows Wu et al. [37] to estimate the model based on a system-generalized method of moments (sys-GMM).

The existing studies use two main approaches to select GMM’s instrumental variables: one is to use official age and tenure as instrumental variables for the replacement of officials; the other is to use lagged variables. Due to the fact that the number of official turnovers is used to measure the level of policy uncertainty, it is not appropriate to use the individual characteristics of an official as the instrumental variable. This paper takes the one-period lagged official turnover variable as the instrumental variable. The estimated results in Table 6 show that the instrumental variable is effective. The impact of policy uncertainty on haze pollution and the spatial spillover effect are still significantly positive after elimination of the endogeneity problem.

This paper tests the robustness of the measurement of policy uncertainty using an indicator variable for officials’ turnover (uncertainty_1) to measure policy uncertainty and re-estimate its impact on haze pollution. It takes the value 1 if there is a change in the party secretary or mayor of a prefecture-level city once within a year and the value 0 otherwise. In addition, given that the local government power is concentrated in local party committees, party secretary turnover (uncertainty_2) is used as an indicator of the level of policy uncertainty. It takes the value 1 if there has been a change in the party secretary in the current year. Table 7 shows the regression results after the change in independent variables, indicating that both the direct effect and the spatial spillover effect of haze pollution aggravated by policy uncertainty are robust.

## 5. Conclusions

This study investigates the impact of policy uncertainty on haze pollution and its spatial spillover effect based on data from prefecture-level cities. The spatial Durbin model is constructed to analyze the effects of policy uncertainty on haze pollution. The research demonstrates that policy uncertainty will exacerbate the local pollution level while significantly spreading over surrounding areas. The effect could be explained by investment expansion, regulatory gaps and responsibility gaps, as a result of policy uncertainty.

Furthermore, geographical distance and economic distance are used to construct intra-provincial and inter-provincial spatial matrices. The matrices assist with our understanding of the potential mechanisms of the spatial spillover effect of haze pollution caused by policy uncertainty. The results show that: if the spatial matrix is constructed based on geographical distance, local policy uncertainty will aggravate haze pollution in the surrounding cities regardless of whether they belong to the same province or not. If the spatial weight matrix is constructed based on economic distance, local policy uncertainty will affect haze pollution only in cities with similar economic distance within the province, but will have no impact on cities in other provinces. The results unveil the dual mechanisms of pollution spatial spillover: one is a geographical mechanism, such as atmospheric circulation, and the other is non-geographical mechanism, such as a race-to-bottom emission among intra-provincial cities with similar levels of economic development.

Last, the paper examines whether central government policies with higher authority can suppress the impact of local policy uncertainty on haze pollution. By examining the implementation of “Measures for the Assessment of the Implementation of the Air Pollution Prevention and Control Action Plan (Trial)”, this paper concludes that central government intervention has a significant positive effect on reducing the impact of policy uncertainty on local haze pollution, but still has limitations in terms of eliminating the direct effect, which is due to the ineradicable nature of policy uncertainty.

The policy implications of the above conclusions are listed below. First, there is a need to systematize the haze control approach to minimize haze policy uncertainty and its implications for both local haze pollution and spatial spillover. At the same time, since policy uncertainty is unavoidable by nature, systematizing the haze control approach will help to abate the positive impact of policy uncertainty on pollution. Second, it is recommended to optimize the performance review system for government officials to avoid the phenomenon of beggar-thy-neighbor emissions. A two-pronged approach could be considered—to strengthen the monitoring of haze pollution along the province borders and to establish a more reasonable performance review system to avoid the race-to-bottom emission competition. Third, policymakers should fully examine the incentive and pragmatic constraints of all parties when they aim to reduce the impact of policy uncertainty. In short, the fundamental solution for curbing the adverse impact of policy uncertainty on haze pollution control is to foster a stable strategy towards haze control within local governments.

## Figures and Tables

**Table 1 ijerph-18-10229-t001:** Moran’s I Index of PM_2.5_: based on distance reciprocal matrix.

Year	Moran’s I	E(I)	Sd(I)	z	*p* Value
2004	0.188	−0.003	0.006	30.656	0.000
2005	0.193	−0.003	0.006	31.418	0.000
2006	0.179	−0.003	0.006	29.251	0.000
2007	0.190	−0.003	0.006	30.925	0.000
2008	0.172	−0.003	0.006	28.190	0.000
2009	0.163	−0.003	0.006	26.661	0.000
2010	0.176	−0.003	0.006	28.735	0.000
2011	0.178	−0.003	0.006	28.999	0.000
2012	0.188	−0.003	0.006	30.706	0.000
2013	0.169	−0.003	0.006	27.566	0.000
2014	0.168	−0.003	0.006	27.498	0.000
2015	0.130	−0.003	0.006	21.469	0.000
2016	0.147	−0.003	0.006	24.027	0.000

**Table 2 ijerph-18-10229-t002:** Results of the impact of policy uncertainty on haze pollution.

	Panel A. Distance Reciprocal Matrix			Panel B. 01 Matrix		
	(1) Direct Effects	(2) Spatial Spillover	(3) Total Effects	(4) Direct Effects	(5) Spatial Spillover	(6) Total Effects
uncertainty	0.018 ***	4.224 **	4.242 **	0.020 **	4.731 **	4.751 **
	(2.83)	(2.45)	(2.45)	(2.43)	(2.12)	(2.12)
lnGDP	−0.019 **	−3.160	−3.179	−0.032 **	−7.389 **	−7.421 **
	(−2.17)	(−1.52)	(−1.52)	(−2.46)	(−2.11)	(−2.11)
lnenv	−0.194	−28.353	−28.548	−0.088	−10.772	−10.860
	(−1.24)	(−0.98)	(−0.98)	(−0.45)	(−0.24)	(−0.24)
lnind	0.005	3.790	3.795	0.035	17.398 **	17.433 **
	(0.30)	(0.94)	(0.94)	(1.24)	(2.27)	(2.27)
lnurban	0.012 **	2.435 *	2.447 *	0.011 **	2.379 *	2.390 *
	(2.46)	(1.95)	(1.95)	(2.23)	(1.80)	(1.81)
lnpden	0.022 *	2.636	2.658	0.032 **	7.277 *	7.309 *
	(1.88)	(1.05)	(1.05)	(2.06)	(1.77)	(1.77)
lnopen	0.006	0.154	0.160	0.003	−0.605	−0.602
	(1.46)	(0.15)	(0.16)	(0.72)	(−0.51)	(−0.51)
lnres	0.408 ***	86.209 **	86.618 **	0.436 ***	100.861 **	101.297 **
	(2.86)	(2.39)	(2.39)	(2.68)	(2.34)	(2.34)
lnmater	0.011 **	2.431	2.443	0.023 ***	5.606 **	5.629 **
	(2.03)	(1.60)	(1.60)	(2.61)	(2.32)	(2.32)
Year/City	YES	YES	YES	YES	YES	YES
Obs	3731	3731	3731	3731	3731	3731

Notes: (1) This table shows the results of the impact of policy uncertainty on haze pollution using the Durbin model. (2) Panel A reports the results using the reciprocal matrix of distance, while Panel B reports the results using the adjacency matrix. (3) Direct effect, spatial spillover and total effect are reported sequentially within each panel. (4) T-statistics are reported in parentheses; *** denotes significance at the 0.01 level; ** denotes significance at the 0.05 level; and * denotes significance at the 0.10 level.

**Table 3 ijerph-18-10229-t003:** Spatial spillover effect: Analysis based on the reciprocal distance matrix.

	Panel A. Intra-Provincial			Panel B. Inter-Provincial		
	(1) Direct Effects	(2) Spatial Spillover	(3) Total Effects	(4) Direct Effects	(5) Spatial Spillover	(6) Total Effects
uncertainty	0.004 **	0.034 ***	0.037 ***	0.012 ***	1.775 **	1.787 **
	(2.06)	(3.02)	(3.05)	(3.66)	(2.51)	(2.52)
lngdp	−0.003	0.017	0.013	−0.011 **	−0.163	−0.173
	(−0.90)	(0.77)	(0.53)	(−1.99)	(−0.17)	(−0.18)
lnenv	−0.041	0.489	0.449	−0.175	−24.776	−24.950
	(−0.38)	(0.71)	(0.59)	(−1.19)	(−1.06)	(−1.06)
lnind	0.008	−0.084 *	−0.076	−0.005	−1.940	−1.945
	(0.80)	(−1.96)	(−1.60)	(−0.41)	(−1.15)	(−1.14)
lnurban	0.003 *	0.018	0.021	0.009 ***	1.768 **	1.778 **
	(1.75)	(1.39)	(1.49)	(2.75)	(2.26)	(2.27)
lnpden	0.016 **	0.030	0.046	0.021 **	2.441 *	2.462 *
	(2.19)	(0.76)	(1.08)	(1.99)	(1.67)	(1.67)
lnopen	0.005 **	−0.013	−0.009	0.001	−1.004 *	−1.003 *
	(2.11)	(−0.97)	(−0.57)	(0.45)	(−1.71)	(−1.71)
lnres	0.084	0.857 **	0.941 **	0.262 ***	30.976 **	31.238 **
	(1.38)	(2.27)	(2.24)	(3.08)	(2.11)	(2.12)
lnmater	0.001	0.007	0.007	0.002	−0.474	−0.472
	(0.58)	(0.80)	(0.80)	(0.91)	(−1.08)	(−1.08)
Year/City	YES	YES	YES	YES	YES	YES
Obs	3731	3731	3731	3731	3731	3731

Notes: (1) This table shows the results of the impact of policy uncertainty on haze pollution using the Durbin model with the reciprocal matrix of distance. (2) Panel A reports the results using the intra-provincial matrix, while Panel B reports the results using the inter-provincial matrix. (3) The direct effect, spatial spillover and total effect are reported sequentially within each panel. (4) T-statistics are reported in parentheses; *** denotes significance at the 0.01 level; ** denotes significance at the 0.05 level; and * denotes significance at the 0.10 level.

**Table 4 ijerph-18-10229-t004:** Spatial spillover effect: Analysis based on the economic distance matrix.

	Panel A. Intra-Provincial			Panel B. Inter-Provincial		
	(1) Direct Effects	(2) Spatial Spillover	(3) Total Effects	(4) Direct Effects	(5) Spatial Spillover	(6) Total Effects
uncertainty	0.004 **	0.011 *	0.015 **	0.006 ***	−0.003	0.003
	(2.00)	(1.89)	(2.14)	(2.81)	(−0.53)	(0.57)
lngdp	−0.007 *	−0.012	−0.019	−0.010 **	0.003	−0.007
	(−1.83)	(−1.02)	(−1.31)	(−2.33)	(0.30)	(−0.66)
lnenv	−0.061	−0.086	−0.148	−0.108	0.518	0.410
	(−0.56)	(−0.24)	(−0.33)	(−0.94)	(1.62)	(1.23)
lnind	0.014	−0.070 ***	−0.057 *	−0.022 *	0.059 **	0.036
	(1.26)	(−2.72)	(−1.91)	(−1.85)	(2.08)	(1.31)
lnurban	0.004 *	0.008	0.011	0.004 *	0.005	0.009
	(1.82)	(1.33)	(1.56)	(1.83)	(0.84)	(1.46)
lnpden	0.017 **	0.005	0.022	0.020 **	0.001	0.021
	(2.08)	(0.23)	(0.83)	(2.13)	(0.06)	(0.98)
lnopen	0.006 ***	0.000	0.006	0.006 ***	0.000	0.006
	(2.61)	(0.02)	(0.67)	(2.67)	(0.10)	(1.23)
lnres	0.155 **	0.907 ***	1.062 ***	0.203 ***	0.213	0.416 **
	(2.48)	(4.64)	(4.44)	(3.01)	(1.17)	(2.17)
lnmater	0.002	0.007	0.009	0.003 *	−0.002	0.001
	(1.42)	(1.37)	(1.50)	(1.76)	(−0.67)	(0.14)
Year/City	YES	YES	YES	YES	YES	YES
Obs	3731	3731	3731	3731	3731	3731

Notes: (1) This table shows the results of the impact of policy uncertainty on haze pollution using the Durbin model with the economic distance matrix. (2) Panel A reports the results using the intra-provincial matrix, while Panel B reports the results using the inter-provincial matrix. (3) The direct effect, spatial spillover and total effect are reported sequentially within each panel. (4) T-statistics are reported in parentheses; *** denotes significance at the 0.05 level; ** denotes significance at the 0.01 level; and * denotes significance at the 0.10 level.

**Table 5 ijerph-18-10229-t005:** Policy uncertainty and haze pollution: The impact of policies at the central level.

	Panel A. before Assessment Measure			Panel B. after Assessment Measure		
	(1) Direct Effects	(2) Spatial Spillover	(3) Total Effects	(4) Direct Effects	(5) Spatial Spillover	(6) Total Effects
uncertainty	0.075 *	2.175 **	2.250 **	0.051 ***	2.175	2.226
	(1.71)	(2.50)	(2.52)	(3.18)	(1.54)	(1.57)
lngdp	−0.007	−0.153	−0.160	0.090 ***	−4.344 **	−4.434 **
	(−1.42)	(−1.44)	(−1.46)	(−7.11)	(−2.23)	(−2.27)
lnenv	−0.051	−2.624	−2.676	1.944 ***	−40.541	−42.485
	(−0.41)	(−1.33)	(−1.30)	(−3.45)	(−1.16)	(−1.20)
lnind	0.030	−0.170	−0.140	0.135 ***	9.008 *	9.143 *
	(1.59)	(−0.55)	(−0.44)	(2.92)	(1.66)	(1.68)
lnurban	0.024 ***	0.164	0.188	−0.010	0.385	0.375
	(2.73)	(0.59)	(0.66)	(−1.07)	(0.70)	(0.67)
lnpden	−0.027	−0.938	−0.964	0.216 ***	3.718 **	3.934 **
	(−0.72)	(−1.22)	(−1.22)	(18.49)	(2.17)	(2.29)
lnopen	0.003	−0.028	−0.025	0.033 *	0.753	0.785
	(1.28)	(−0.49)	(−0.42)	(1.75)	(0.86)	(0.88)
lnres	0.232 *	2.869	3.101	0.239 ***	14.688 **	14.927 **
	(1.74)	(1.14)	(1.19)	(2.60)	(2.15)	(2.17)
lnmater	−0.003	−0.048	−0.052	0.046 ***	2.300 *	2.347 *
	(−0.96)	(−0.56)	(−0.58)	(3.65)	(1.79)	(1.81)
Year/City	YES	YES	YES	YES	YES	YES
Obs	2810	2810	2810	921	921	921

Notes: (1) This table shows the results of the impact of policy uncertainty on haze pollution using the Durbin model. (2) Panel A reports the results using the sample before the introduction of a strict and unified assessment measure, while Panel B reports the results using the sample after the introduction of the strict and unified assessment measure. (3) The direct effect, spatial spillover and total effect are reported sequentially within each panel. (4) T-statistics are reported in parentheses; *** denotes significance at the 0.01 level; ** denotes significance at the 0.05 level; and * denotes significance at the 0.10 level.

**Table 6 ijerph-18-10229-t006:** System GMM model estimation results.

	ln PM_2.5_
uncertainty	0.306 ***
	(2.77)
turnover × W	0.212 ***
	(3.34)
Constant	−2.099 **
	(−2.32)
Obs	3731
R2	0.219
F	49.807
LM test	49.041 ***

Notes: *** denotes significance at the 0.01 level; ** denotes significance at the 0.05 level.

**Table 7 ijerph-18-10229-t007:** Robustness test.

	Panel A. Officials’ Turnover			Panel B. Secretary of Municipal Party Committee’s Turnover		
	(1) Direct Effects	(2) Spatial Spillover	(3) Total Effects	(4) Direct Effects	(5) Spatial Spillover	(6) Total Effects
uncertainty_1	0.031 **	1.545 *	1.576 *			
	(2.53)	(1.80)	(1.82)			
uncertainty_2				0.045 ***	2.129 *	2.174 *
				(2.68)	(1.79)	(1.81)
lngdp	−0.082 ***	−4.223 ***	−4.305 ***	−0.082 ***	−4.314 ***	4.396 ***
	(−9.08)	(−3.43)	(−3.47)	(−9.00)	(−3.40)	(−3.44)
lnenv	1.700 ***	39.133 *	40.833 *	1.706 ***	40.302 *	42.008 *
	(4.49)	(1.77)	(1.82)	(4.48)	(1.77)	(1.82)
lnind	0.090 ***	7.605 **	7.694 **	0.092 ***	7.880 **	7.972 **
	(2.72)	(2.47)	(2.48)	(2.76)	(2.46)	(2.47)
lnurban	−0.008	0.494	0.486	−0.008	0.525	0.517
	(−1.07)	(1.11)	(1.08)	(−1.02)	(1.14)	(1.11)
lnpden	0.215 ***	3.816 ***	4.030 ***	0.215 ***	3.907 ***	4.122 ***
	(24.73)	(3.35)	(3.52)	(24.55)	(3.32)	(3.49)
lnopen	0.025 **	0.621	0.645	0.026 **	0.695	0.720
	(2.42)	(1.08)	(1.11)	(2.50)	(1.17)	(1.19)
lnres	0.260 ***	15.566 ***	15.826 ***	0.261 ***	15.894 ***	16.155 ***
	(3.63)	(3.14)	(3.17)	(3.63)	(3.11)	(3.14)
lnmater	0.031 ***	1.734 **	1.765 **	0.032 ***	1.782 **	1.814 **
	(3.92)	(2.34)	(2.36)	(3.92)	(2.33)	(2.35)
Year/City	YES	YES	YES	YES	YES	YES
Obs	2810	2810	2810	921	921	921

Notes: (1) This table shows the results of the impact of policy uncertainty on haze pollution using the Durbin model. (2) Panel A reports the results using official turnover as a proxy for uncertainty, while Panel B reports the results using the secretary of municipal party committee’s turnover. (3) The direct effect, spatial spillover and total effect are reported sequentially within each panel. (4) T-statistics are reported in parentheses; *** denotes significance at the 0.01 level; ** denotes significance at the 0.05 level; and * denotes significance at the 0.10 level.

## Data Availability

The published statistical data comes from the website (https://sedac.ciesin.columbia.edu/) (accessed on 20 September 2020).

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
