# Peer review of "Will Policy Uncertainty Deteriorate Haze Pollution? A Spatial Spillover Perspective"

_ijerph, 2021, doi:10.3390/ijerph181910229_

Round 1

Reviewer 1 Report

The work is interesting and well conceived. It is necessary to write in the third person singular.

Author Response

Dear Reviewer:

Thank you for the wonderful comments concerning our manuscript entitled “Will Policy Uncertainty Deteriorate Haze Pollution? ——Based on the perspective of Spatial Spillover” (ID: ijerph-1316348). Those constructive comments are all valuable and very helpful for revising and improving our paper. We have studied your comments carefully and have made revisions which we hope meet with your approval. Please see the attachment for our response.

Best regards.

Reviewer 2 Report

The concept behind the paper has some merit but is very poorly presented. The paper reads more like an extended report rather than a scientific paper. The paper needs a more concise presentation framework with clear statement of the objective.

One of the underlying assumptions appears to be that the input data used for variable formulation have littler or no uncertainty. This is a problem as uncertainty is present in all data sets and cannot be ignored. There needs to be input from atmospheric scientists as well as meteorologists to get the underlying science presented more clearly.

One point which needs to come across is that the decentralized system of governance is not working very well and this is  particularly the case with respect to environmental protection and coordination at the local and regional levels. The paper needs to be more forthright on this issue and how to fix it. The apparent lack of continuity in scientific knowledge and understanding due to frequent personnel turnover due to conflicts of interest and thus lack of the needed commitment by personnel at the local and regional levels is one example.

Author Response

(The authors gave the same response as above.)

Reviewer 3 Report

Comments to authors

The authors aimed to assess the direct and spillover effects of policy uncertainty on haze pollution at the prefecture level in China. This is an active research area and would be interesting to readers of IJERPH. To reach this goal, they used a spatial Durban model to access the spatial spillover and impact of the assessment measures and policies. 

I enjoyed reading this article. Your work is valuable and can be better highlighted by (1) careful review of the grammar and phrasing of the manuscript (2) modifications and suggested additions to the manuscript, especially around the justifications of your methods selection. 

Specific comments:

  1. Please review the manuscript. Many areas of the manuscript could be reviewed for phrasing. For example, in the introduction, the phrase “haze pollution is still very enormous” is a bit informal. Something like “haze pollution is a widespread and major issue in China, even after the introduction of policies to address it” might be better. 
  2. I like your introduction overall. You may wish to add in detail on the relationship between haze and the structure of industry. The relationship between industrial restructuring haze may modify the relationship you are looking at, so it is important to at least conceptually consider - see Chen et al (2019). 
    1. Chen, S., Zhang, Y., Zhang, Y., & Liu, Z. (2019). The relationship between industrial restructuring and China's regional haze pollution: A spatial spillover perspective. Journal of Cleaner Production, 239, 115808.
  3. You may also wish to look into the modeling approach used in Chen et al. (2019). At least justify why the spatial Durban model approach is more appropriate than the semi-parametric global vector autoregressive model approach. 
  4. In Table 5, would the “Report on the State of the Environment in China” in 2015 have any effect? You may wish to add an additional break point.

Author Response

(The authors gave the same response as above.)

Round 2

Reviewer 2 Report

Review comments are attached.

Author Response

Dear Reviewer:
Thank you for your letter and for the wonderful comments concerning our manuscript entitled "Will Policy Uncertainty Deteriorate Haze Pollution? ——Based on the perspective of Spatial Spillover" (ID: ijerph-1316348). Those constructive comments are all valuable and very helpful for revising and improving our paper. We have studied your comments carefully and have made revisions which we hope meet with your approval.
